# How Did COVID-19 Pandemic Stress Affect Poles' Views on the Role of the Forest?

Emilia Janeczko [1] , Jan Banaś [2] , Małgorzata Woźnicka [1] , Krzysztof Janeczko [1], Katarzyna Utnik-Banaś [3] , Stanisław Zięba [2] and Jitka Fialova [4],*

1 Institute of Forest Sciences, Warsaw University of Life Sciences—WULS, Nowoursynowska 159, 02-776 Warsaw, Poland; emilia_janeczko@sggw.edu.pl (E.J.); malgorzata_woznicka@sggw.edu.pl (M.W.); krzysztof_janeczko@sggw.edu.pl (K.J.)

2 Department of Forest Resources Management, University of Agriculture in Kraków, al. 29 Listopada 46, 31-425 Kraków, Poland; jan.banas@urk.edu.pl (J.B.); stanislaw.zieba@urk.edu.pl (S.Z.)

3 Department of Management and Economics of Enterprises, University of Agriculture in Kraków, al. Mickiewicza 21, 31-120 Kraków, Poland; katarzyna.utnik-banas@urk.edu.pl

4 Department of Landscape Management, Faculty of Forestry and Wood Technology, Mendel University in Brno, Zemědělská 1, 613 00 Brno, Czech Republic

* Correspondence: jitka.fialova@mendelu.cz

**Abstract:** The 2020–2021 COVID-19 pandemic has had a tremendous impact on the daily lives of everyone, including local communities and entire societies. Under the influence of this new experience, the importance of the services and benefits provided by forests and other green spaces has increased. A very large role in this aspect was played by media messages promoting the idea of being close to nature as a remedy for malaise and stress, and pushing the thesis that the risk of spreading the virus outdoors is lower than it is indoors. Thanks to media messages, as well as government responses (i.e., lockdown, temporary bans on entering the forest), public attention has been directed toward forests, generating greater interest in forest management and conservation issues, as well as in nature and forestry education. The purpose of our research was to determine how the pandemic affected the frequency of visits to the forest and how it changed the public's views on the role of forests. The research material consists of the results of a questionnaire survey (online and traditionally way) carried out in Poland from September to October in 2020. A total of 1402 people were surveyed. The results show that nearly 52% of respondents increased their use of forest recreational services during the COVID-19 pandemic. We also found that more than 80% of respondents agreed with statements that the forest is a safer space than, for example, parks or squares, and it is more difficult to contract the virus there. Men were more likely to agree with this statement than women (1.51), respondents without children (1.45), respondents over the age of 31 (1.72), and respondents with more than primary or secondary education (1.37). Also, more than 80% of respondents said that the social functions of the forest (e.g., recreational) had gained importance as a result of the pandemic. The social functions of the forest gained importance primarily among respondents with higher education (2.40), and among respondents who had visited the forest rather infrequently (several times a year) for recreational purposes before the pandemic (1.72). Those with children were more likely to agree with the statement that the economic functions of the forest have lost their importance (1.43), as were those who had formerly visited the forest several times a year (1.53). With regard to the statement "the slowdown of the economy has contributed to the improvement of the environment," there were no statistically significant differences in the views of respondents in terms of their socio-demographic characteristics.

**Keywords:** pandemic; forests; ecosystem services; recreation

## 1. Introduction

The global state of the COVID-19 pandemic was declared by the World Health Organization on 11 March 2020. This news was surprising to both citizens and governments

of many countries. The need for social distance, lockdown, and home isolation, and the consequent restriction of outdoor leisure activities, left a strong mark on the lives of millions of people [1,2]. The 2020–2021 COVID-19 pandemic changed the lifestyles of local communities and entire societies [3], causing numerous anomalies in people's daily lives, including work, education and recreation, as well as impacting overall mobility and physical activity. The pandemic has made many people realize the importance of nature, including forests, in everyday life. Many people have started to realize what services and benefits forests and other green areas provide us with. The widespread media message that the risk of the virus spreading outdoors is lower than indoors was important. Outdoor spaces allow for airflow, ventilation, and lack of air recirculation, and allow for greater physical distance, which reduces the risk of virus transmission through larger respiratory droplets [4]. In addition, outdoor environments also have fewer touch surfaces that can harbor the virus. Ultraviolet light, present outdoors due to sunlight, causes a 10-fold decrease in virus survival on surfaces [5]. At the same time, there has been an increasing number of scientific reports in the media space about the beneficial effects of contact with nature on human health. Scientific evidence indicates that even short visits to natural sites help to restore and sustain mental well-being [6–8] as well as reduce stress levels [9–11]. Son and Ha [10] argued that increasing contact with nature helps to improve social and emotional interactions in modern society. Bang et al. [12] proved that physical activity in the forest has a positive effect on self-esteem and social interaction for children. Previous studies have shown that people's moods and positive emotions improve in natural environments [6,8,13,14]. Governments of individual countries during the pandemic took various measures to limit the spread of coronavirus. In Poland, such measures included restrictions on movement, access to services, and a short-term ban on entering the forest. During the first lockdown periods, public access to outdoor recreational facilities was restricted in many countries to reduce contact with other people while preventing the spread of infectious diseases [15,16]. During the COVID-19 pandemic, public access to forest areas was the subject of considerable national and international discussion and changed during the pandemic period around the world [3,17–19]. In Poland, from 3 April to 11 April 2020, there was a ban on entering forests under the management of the State Forests (a company that manages about 80% of Poland's public forests) and all national parks. The restriction was dictated by concern for the health and safety of people, and was a direct result of two reasons related to the state of the epidemic—the speed and ease of the spread of the virus in conditions of close contact between people, and the fact that, despite requests and appeals, there were still many people using the time of isolation for picnics and social gatherings, among others reasons, in areas managed by the State Forests.

However, this decision was very quickly challenged by, among others, the Ombudsman, as having no legal basis and constituting an interference with everyone's right to use the environment and the constitutional right to move freely in the country. Paradoxically, the ban on access to forests has caused public attention to be directed toward forests, thus triggering greater interest in forest management, nature conservation, and nature–forest education issues. Hence, the purpose of our research was to determine how the pandemic affected the frequency of visits to the forest and how it changed public views on the role of forests. Achieving this goal involved the need for two more specific objectives. One was to determine whether people were more likely to visit forests for recreation as a result of the pandemic. The second specific objective was to show whether societal views about forests, in particular the social functions of the forest, changed under the stress of COVID-19. We adopted two hypotheses, namely:

- The COVID-19 pandemic contributed to increased recreational use of the forest.
- The increase in the frequency of visits to the forest caused by the COVID-19 pandemic has increased the perceived importance of the social function of the forest, thereby increasing the importance of the cultural services of the forest.

## 2. Materials and Methods

### 2.1. The Questionnaire

The research material consists of the results of a questionnaire survey carried out in Poland from September to October in 2020; so, during the period of relaxed restrictions, between the first and second waves of the coronavirus epidemic (the first lockdown lasted from March to May; the second lockdown lasted, in Poland, from 24 October to 29 November 2020, followed by a period of slowly easing restrictions). Due to the ongoing pandemic, we decided to conduct an online survey. A link to the survey, created on the Webankieta platform 2020, was made available on social networks such as Facebook (Meta Platforms, Inc., Meta, CA, USA) Instagram (Meta Platforms, Inc., Meta, CA, USA), and Twitter, as well as on the website of the individual forest districts comprising the 17 directorates of the State Forests, namely, Radom RDSF. Each forest district had a so-called 'liaison officer', usually an employee of the district, who, on his/her days off, outside of his/her professional duties, forwarded the link to the survey to all persons interested in forest recreation in the region. Liaison duties were largely responsible for the selection of respondents. Their task was to forward the survey link to people they met in the forest, using recreational facilities, walking in the forest, etc., as well as to people who contacted the forest districts about educational activities or anything else related to their stay in the forest. Each time, liaison duties asked about the age of the respondents and the purpose of their visit. The questionnaire was completed only by adults over the age of 18 who were interested in recreation in the forest. In the questions we emphasized that we wanted answers on issues relating to these specific forests and the region in which we were conducting our research.

The survey questionnaire included questions about gender (female, male), age (18–30 years, 31–40 years, 41–50 years, and above 50 years), level of education (primary, secondary, and higher), place of residence (rural area, small town with up to 15,000 inhabitants, medium town with 15,000–100,000 inhabitants, and large town with more than 100,000 inhabitants), satisfaction with their standard of living a as well as the frequency of recreation in the forest (every day, several times a week, several times a year).The questionnaire comprised 19 thematic questions relating to various issues related to the ecosystem functions of the forest. Most of them were questions with a Likert scale of five degrees of appreciation (very important, important, moderately important, not very important, and irrelevant) through which we could obtain knowledge about the degree of acceptance of the analyzed phenomena, views, processes, features of the forest, etc. This article draws on respondents' views on the following issues:

How did your frequency of visits to the forest change during the pandemic (decisively increased, increased, no change, decreased, or decisively decreased)?

Views on the following media messages: It is more difficult to become infected with the virus in a forest; the forest is a safer space than other green areas (parks, squares, etc.); the social functions of a forest (e.g., recreation) gained importance during a pandemic; the economic functions of a forest (e.g., timber harvesting) lost importance; the slowdown of the economy contributed to the improvement of the natural environment in Poland.

Respondents were asked to assess each report using one of five selectable responses ordered on a Likert scale.

### 2.2. Participants

A total of 1402 respondents took part in the survey, including 655 women (46.72%) and 747 men (53.28%). Respondents aged 31–40 years (31.1%) comprised the most numerous group. Respondents aged 18–30 accounted for 27.82% of respondents, followed by those aged 41–50 (23.97%) and those above 51 years (17.12%). Rural residents comprised the most numerous group (42.44%). The survey included 807 urban residents (57.56%), of whom 18.54% of respondents were from small cities (up to 15,000 residents), 23.18% from medium-sized cities (15–100,000 residents), and 15.83% from large cities (over 100,000 residents). The vast majority of respondents had a higher education (64.55%). Secondary education was held by 32.03% of respondents and primary education by only 3.42%. The majority of

respondents (63.86%) found their material standard of living satisfactory, 21.81% found it not very satisfactory, 9.69% found it fully satisfactory, and 4.63% found it unsatisfactory.

Almost half of the respondents (49.61%) declared that they rest in the forest several times a week; 37.7%, several times a year; and 12.69%, every day.

### 2.3. Statistical Analysis

In the first step, the frequency of respondents' selected answers was determined on a five-degree Likert scale. The third answer "has not changed" was taken as neutral representing a comparative level. The results were presented both for individual levels and in aggregate form combining the first response with the second (definitely decreased and decreased) representing decreasing influence, and the fourth with the fifth (increased and definitely increased) reflecting increasing influence.

A logistic regression (LR) model was used to determine the detailed sociodemographic profile of individuals in terms of their perceptions of the pandemic's impact on the phenomena under study. This model determines the probability that the dependent variable (here, the third and fourth answers, agree and strongly agree) will take the value of 1, provided that the explanatory variables (x1, x2, . . ., xi) take certain values [20]. A logistic model with one explanatory variable is defined by the following formula:

$$P(x) = (\exp(\beta 0 + \beta 1\ x1))/(1 + \exp(\beta 0 + \beta 1\ x1)) \tag{1}$$

If Equation (1) is subjected to a logit transformation, we obtain the logit form of the model, which, generalized to multiple variables, is represented by the following equation:

$$logitP = \beta_0 + \sum_{i=1}^{k} \beta_i x_i \tag{2}$$

The logit form of the model shown by Equation (3) is commonly used in research because of the intuitive, simple interpretation of the right-hand side of the equation as a linear function.

Sociodemographic characteristics of respondents were used as potential explanatory variables. All variables were binary in nature, taking the value 1 if the trait applied to the respondent and 0 if the respondent did not have it.

The trait "gender" was described by two variables: "female" (F) and "male" (M). If the respondent was female, the variable F = 1, otherwise F = 0. The variable M was coded in the same way. For variables consisting of several categories, additional variables were introduced. For example, the age variable included four age ranges coded with the following variables: respondents aged 18 to 30, A; respondents aged 31 to 40, B; respondents aged 41 to 50, C; respondents aged over 50, D. In addition, we used the BCD (>30 years old) and CD (>40 years old) categories. Correlations between the preliminary independent variables were tested, but ultimately no such significant relationships (*p*-value < 0.05) were found between the proposed explanatory variables.

A stepwise approach (progressive regression) was used to build a multivariate model. In the first step, a model with one independent variable was developed and checked to see if it was significantly different from a model containing only a fixed member. In subsequent steps, additional variables were added and their significance was assessed; if the variables proved to be insignificant, they were removed from the model. The quasi-Newton method was used to parameterize the model. The significance of the model parameters was assessed using the Wald test. The goodness of fit of the model to empirical data was assessed using the Hosmer–Lemeshow test [21].

The modeled probability P(x) that a respondent agreed with the studied opinion was determined by substituting, into Equation (2), the corresponding values of the explanatory variables (1 or 0) describing the respondent's sociocultural characteristics.

In the logistic regression model, in addition to the regression coefficients and their statistical significance, another important parameter is the odds ratio (OR). The OR is the ratio of the chance S(A) of an event occurring in group A (e.g., women agree with a

given opinion) to the chance S(B) of that event occurring in group B (men agree with a given  opinion).

$$OR_{AxB} = \frac{S(A)}{S(B)} \qquad (3)$$

OR = 1 indicates that the chance of agreeing with a given opinion is the same in the women's group as in the men's group; OR > 1 indicates that, in the first group (women), agreement with a given opinion is significantly higher than it is in the second group (men). Conversely, OR < 1 indicates that, in the first group (women), agreement with a given opinion is lower than it is in the second group (men).

For the logit form of the model with multiple variables, we used the following formula to determine the odds ratio:

$$OR_{AxB} = e^{\sum_{j=1}^{k}(X_{Aj}-X_{Bj})\beta_j} \qquad (4)$$

## 3. Results

The majority of respondents (51.92%) (Figure 1) said that the pandemic caused them to visit the forests more often. A fairly high percentage of respondents (39.94%) said that the frequency of their visits to the forest during the pandemic did not change. In contrast, about 8% of respondents said they visited forests less often as a result of the pandemic. It was observed that an increase in visits to the forest was more often declared by residents of small- and medium-sized cities, as well as those living in rural areas. Respondents with secondary and higher educations visited forests more frequently during the pandemic. Also, those who visited forests relatively infrequently before the pandemic (several times a year) and those who went to the forest daily did not change the frequency of their visits to the forest during the pandemic.

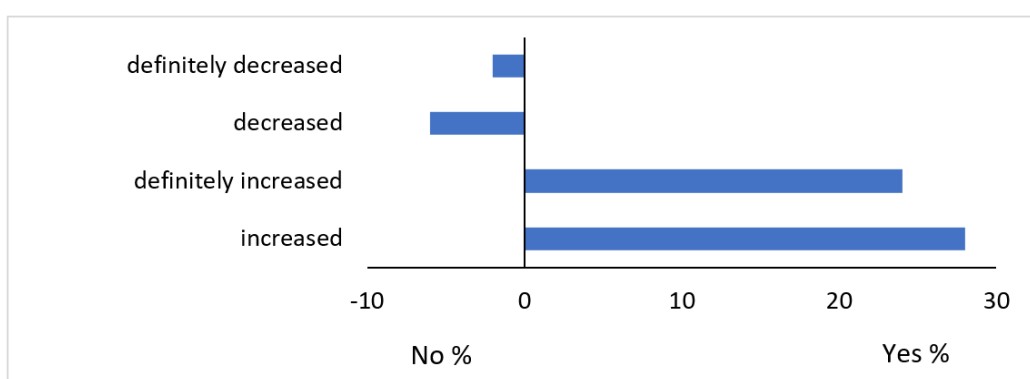

**Figure 1.** Respondents' opinions on the impact of the pandemic on the frequency of visits to the  forest.

The vast majority of respondents (more than 80% of respondents each time) agreed with statements such as: "it is more difficult to contract the virus in the forest"; "the forest is a safer space than public green spaces such as parks or squares"; the social functions of the forest (e.g., recreation) have gained in importance as a result of the pandemic" (Figure 2). For the other two statements, it was found that a fairly significant percentage of respondents (about 30% each time) could not clearly answer whether the economic functions of the forest had lost importance as a result of the pandemic, or whether the slowdown in the economy had contributed to the improvement of the natural environment. Respondents' views varied by socio-demographic characteristics.

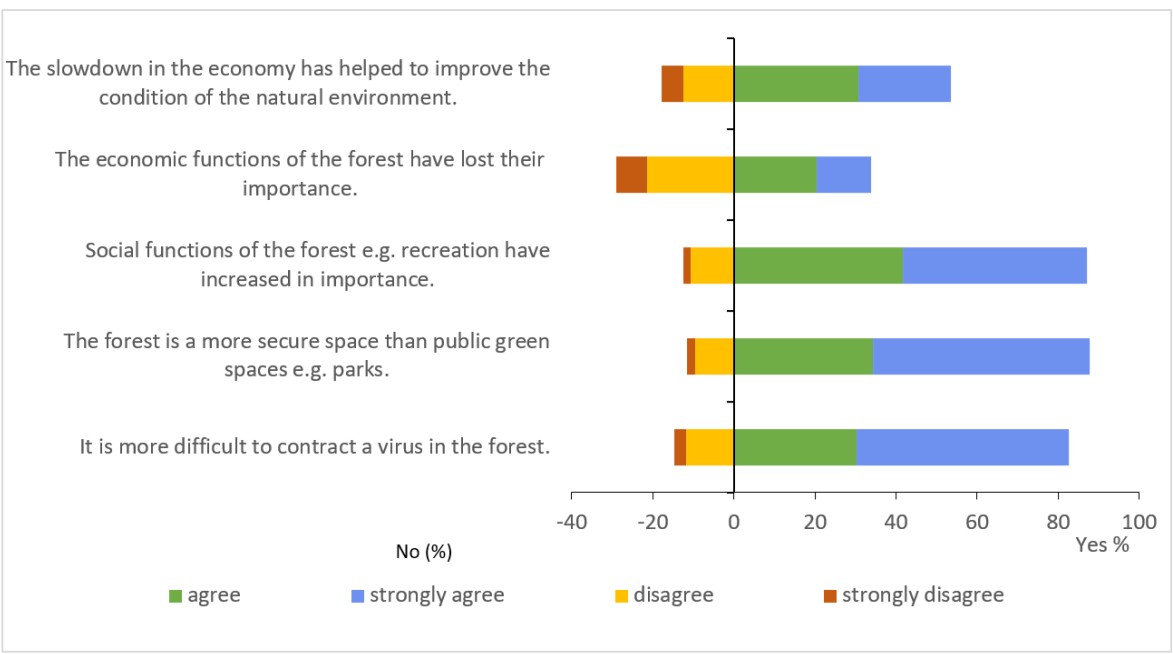

**Figure 2.** Respondents' opinions on the importance of the forest during the pandemic period.

Respondents' opinions on whether it was more difficult to contract the virus in the forest varied by gender, family situation, age, education, and place of residence of respondents (Table 1, Figure 3). Men were more likely than women to agree with this statement. Respondents without children were more likely to answer affirmatively than those who were parents. Respondents over the age of 31 were also more likely to agree with this statement than younger respondents. Similarly, respondents with higher education were also more likely to agree with the statement than those with primary or secondary education. And interestingly, urban residents were less likely than rural residents to agree with this statement.

Respondents with higher education were more likely to agree with the opinion that the forest is a safer space than public green spaces, compared to respondents with primary or secondary education. More often, a similar view was expressed by those who declared a satisfactory level of material life, as well as by those who reported going to the forest several times a year for recreation and leisure.

Respondents with a university education were more likely to express the opinion that the social functions of the forest had gained importance during the pandemic, compared to those with primary or secondary education. Similarly, those respondents who usually visited forests several times a year for recreational purposes were also more likely to respond affirmatively to the question regarding the increased importance of the forest's social functions as a result of the pandemic.

The statement that the economic functions of the forest have lost importance was more often agreed with by those who had children, as well as by those who usually visited forests several times a year. Only with regard to the statement "the slowing down of the economy has helped improve the condition of the natural environment" were no statistically significant differences observed in the views of respondents with regard to their socio-demographic characteristics.

**Table 1.** Logit models of perceptions of the role of the forest during the pandemic period.

| Opinion | Intercept | Gender (1) | Age (2) | Education (3) | Residence (4) | Children (5) | Financial Status (6) | Frequency of Forest Visiting (7) |
|---|---|---|---|---|---|---|---|---|
| | | | | Variable, Coefficient, (Wald's χ2) | | | | |
| It is more difficult to contract a virus in the forest | 1.029 *** (39.05) | M, 0.415 ** (8.22) | A, −0.674 *** (12.27) | U, 0.314 * (4.46) | V, 0.514 *** (11.55) | N, 0.372 * (3.98) | ns | |
| The forest is a more secure space than public green spaces, e.g., parks. | 1.001 *** (31.02) | ns | ns | U 0.646 *** (14.54) | | | S 0.504 ** (8.21) | >t 0.430 ** (6.65) |
| Social functions of the forest, e.g., recreation have increased in importance. | 1.088 *** (55.45) | ns | ns | U 0.874 *** (29.23) | | | | >t 0.545 *** (11.30) |
| The slowdown in the economy has helped to improve the condition of the natural environment. | −1.165 *** (85.85) | | | | | Y 0.355 ** (8.85) | | >t 0.424 *** (12.54) |

Significance level: * $p < 0.05$, ** $p < 0.01$, *** $p < 0.001$, ns—variable not significant; (1) M—male; (2) A—18–30 years old; (3) U—university; (4) V—village; (5) N—without children, Y—with children; (6) S—satisfied, (7) less than once a week.

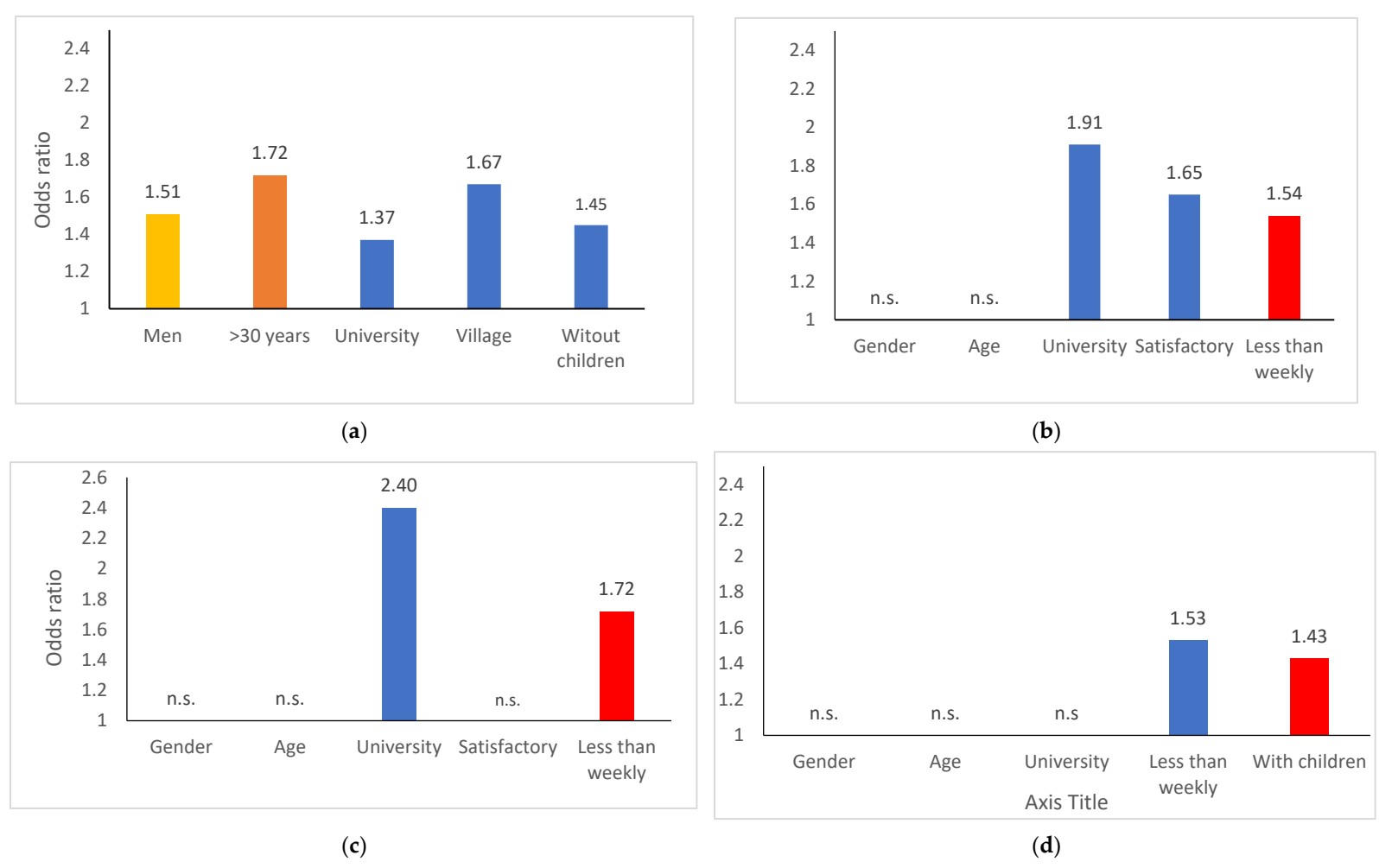

**Figure 3.** Influence of respondents' socio-demographic characteristics on the role of forests during the pandemic: (**a**) the forest is more difficult to contract the virus, (**b**) the forest is a safer space than parks, (**c**) social functions have gained importance, (**d**) economic functions have lost importance. n.s.—not statistically significant.

## 4. Discussion

From our study, it is clear that the frequency of visits to the forest during the pandemic increased.

There are a number of papers [17,19,22,23] indicating that the frequency of visits to forests and outdoor recreational activities in green areas, including suburban forests, protected areas, and urban parks, increased after the COVID-19 blockade. Also, the study of Grima et al. [24] shows that the frequency of visits to natural areas and urban forests increased significantly during the pandemic, and that the importance of these areas increased significantly. A comparison of the number and distribution of forest visitors before and during the COVID-19 blockade in Bonn, Germany, showed that there was a significant increase in forest visitation [17]. Similarly, Geng et al. [25] found that the frequency of visits to urban parks was significantly higher during COVID-19 compared to pre-pandemic baseline levels. A study by Ciesielski et al. [26] conducted in Poland found that observed changes in recreational use in surveyed forest areas during the pandemic compared to the previous year varied by pandemic period and study area. The ban on access to forest areas significantly reduced the number of visits to forests in all study areas. However, the number of visits to suburban forests and remote natural tourist sites increased during the later periods of the pandemic, especially during the summer months of 2020. On the other hand, however, there are also studies that show that participation in visits to urban parks and green spaces, outdoor activities in forests and mountains, and trips to mountain villages decreased after the outbreak of the COVID-19 pandemic [27]. Park et al. [28] argues that the number of respondents visiting urban green spaces and forests was lower during the COVID-19 pandemic than before the COVID-19 pandemic. Our findings on the relationship between socio-demographic characteristics and frequency of forest visits provide a better understanding of people's behavior during the pandemic. They demonstrate that pandemic stress was at a higher rate for residents of large cities, who generally limited their visits to forests out of concern for their health. Also, Chang et al. [27] found that residents of the Seoul metropolitan area decreased their participation in forest recreational activities compared to residents outside the area. Chang et al. [27] found, furthermore, that people over the age of 40 were more likely to reduce their participation in all types of forest recreational activities than people in their 20s. However, our study did not show this. Instead, we found that respondents with secondary and higher educations were more likely to visit forests during the pandemic, which we associate with the fact that environmental awareness, as well as understanding of the impact of contact with nature on health, increases with educational level. Our research shows that people were eager to visit forests, being convinced that it is more difficult to contract a virus in the forest. This belief was not so much intuitive, but rather dictated by media messages. Our observations on the frequency of visits to the forest are reflected in findings on the influence of socio-demographic characteristics on public opinion about the role of the forest during a pandemic. Shammi et al. [29] found that the COVID-19 pandemic had a more significant impact on vulnerable populations, such as young children and the elderly. Using our study as an example, it is clear that the stress of COVID-19 greatly affected people living in cities who had a primary education, but also greatly affected women and young people, especially those with children. These were the very groups far more cautious about the claim that it was harder to contract the virus in the forest, and less likely to identify with it. Insecurity, difficulties in understanding, and human adaptation to the changes taking place are challenges today that breed stress, uncertainty about tomorrow, and isolation from society [30,31]. The pandemic has undoubtedly heightened feelings of fear and had a decisive impact on the perception of risk of contracting the virus. Shepherd et al. [32], Barke et al. [33], Davidson et al. [34], and Flynn et al. [35] found that risk perception is influenced by a number of factors such as gender, age, and location. These same factors determined respondents' view of the role of the forest during a pandemic. Barke et al. [33] believe that one of the most consistent findings from studies of people's risk perceptions is that women express much greater concern than men about health and environmental

risks. Feingold [36] and Randler et al. [37] showed that women are more anxious than men on the general trait of anxiety or neuroticism. Slimak and Dietz [38] found that better-educated and more affluent people are less concerned about elements of environmental risk, perhaps this also relates to the risk of coronavirus spread. Open areas are able to provide urban residents with healthy and safe outdoor recreation areas and green infrastructure. According to Chen at al. [39], the increased popularity of this view is clearly indicated by the increased frequency of visits to forests and urban parks. Parks provide environmental, social, psychological, and health functions and ecological services to residents. Forests have gained renewed attention for their vital and irreplaceable functions, which have been clearly beneficial to human public health as well as social welfare during the health crisis and global viral pandemic. In many countries, the difference between parks and urban forests is blurring. In Poland, it is different. Parks are characterized by a much greater degree of composed greenery (not just trees, but plantings of ornamental shrubs, grasses, and perennials) and, above all, a greater saturation of recreational infrastructure; thus, there are more touch surfaces that could have harbored the virus during the pandemic. Also, in general, the frequency of users of urban parks is much higher than that of urban or suburban forests. In our research, we wanted to determine to what extent people recognize this fact and understand that a forest is safer than a typical urban park in terms of the possibility of contracting coronavirus. It turns out that respondents with higher education were more likely to agree with this opinion compared to respondents with primary or secondary education. More often, a similar view was expressed by those who declared a satisfactory level of material life, as well as by those who went to the forest several times a year for recreation and leisure. We did not come across studies that compared the number of visits to city parks with, separately, the number of visits to forests during the pandemic period. Hence, we cannot relate the results obtained to the frequency of visits to open areas during the pandemic period. We think that in this case, the key factor may have been the knowledge of the respondents, which is generally coupled with higher education, and thus also, generally, with higher material living standards. Our research has unequivocally shown that the social functions of the forest have become more important as a result of the pandemic. This is confirmed by many other studies. Lopez et al. [40] found that New York City residents considered the role of urban green spaces in mental and physical health to be more important during the pandemic than before. Also, a study by Beckmann-Wübbelt et al. [41] found that the value of urban forest cultural services during the pandemic, especially for recreational services in suburban forests, increased significantly. Park et al.'s [28] study found that the cultural services of forest ecosystems, such as forest landscape, recreation, and therapy, were considered significantly more important during the COVID-19 pandemic. Individuals who experienced socio-psychological stress due to reduced recreational activities in the forest increased their perceived importance of cultural services, including of forest therapy. Also, the results of a study by Pichlerová et al. [18] showed that the pandemic strengthened perceptions of forests as high-quality regenerative environments, and the emotions associated with visits to the forest played an important role in perceptions of the social importance of forests and their possible overexploitation. On the other hand, however, it should be noted that even before the pandemic, an increase in the importance of the social functions of the forest was observed in many countries, especially those within the reach of large cities [12,42,43]. There has been considerable and increasing global attention towards using forest environments as places for recreation and health promotion [44]. Also, Patel et al. [45], Tarrant and Cordell [46], Blazevska et al. [47], and Pichlerova et al. [18] argue that demand for non-market forest values, such as aesthetic, cultural, spiritual, and recreational values, has been increasing in recent decades. We found that the social functions of the forest gained importance during the pandemic primarily in the eyes of people with higher education, as well as by those who had visited forests quite rarely (several times a year) before the pandemic. This is somewhat confirmed by Koprowicz et al.'s [48] study, which found that those with higher education were most likely

to admit the benefits of spending time in the forest, while those with primary education were least likely to do so.

The view that the importance of the social function of the forest increased during the pandemic is not equivalent to the view that the economic function of the forest decreased during the same period. Our research showed that a large group of respondents (almost 1/3) could not make a clear statement on this issue. This may be a result of the public's generally not very high knowledge of economics. The "Survey of Poles' Economic Awareness and Knowledge 2020" report conducted by the National Bank of Poland (accessed on 2 January 2024, https://nbp.pl/edukacja/badania-wiedzy-ekonomicznej/) [49] showed that objective knowledge was low or very low for 14% of Poles, medium for 33%, and high or very high for 45% and 7%, respectively. At the same time, the self-assessment of Poles' economic knowledge was also quite low. The statement that the economic functions of the forest have lost their importance was more often agreed with by those who had children, as well as by those who usually visited forests several times a year. For families with children, recreation in the forest is an opportunity to spend leisure time together, build strong family ties, experience nature, and be physically active. Recreation in the woods does not involve the need for special skills, and does not require specialized equipment or large financial outlays, which is especially important for less affluent families. Families with children are a social group that values the social functions of the forest more than the economic functions. During the period of the pandemic, there were messages in the media indicating that the pandemic brought enormous socioeconomic losses, but also benefits to the natural environment. Lenzen et al. [50] claim that environmentally, the pandemic brought a reduction in greenhouse gas emissions and air pollution, mainly due to a decrease in fossil fuel consumption. Rume et al. [51] points out that the pandemic situation has significantly improved air quality in various cities around the world, reduced greenhouse gas emissions, decreased water and noise pollution, and reduced pressure on tourist sites, all of which can help restore the ecological system. On the other hand, there have also been some negative consequences of COVID-19, such as the increase in medical waste; the haphazard use and disposal of disinfectants, masks, and gloves; and the burden of untreated waste that constantly threatens the environment. Our research shows that although the majority of respondents agree with the statement that the slowdown in the economy due to the pandemic has contributed to the improvement of the environment, almost 1/3 of respondents could not clearly comment on this issue. We think that this is largely related to the general level of environmental awareness of Polish residents, which Stefaniuk [52] says, and which was confirmed, is not optimal but is gradually increasing.

## 5. Conclusions

The pandemic has caused a change in views on many issues concerning the natural environment and human well-being. In Poland, forests occupy almost 1/3 of the country's land area, and public forests dominate, which makes them an important component of space and an area of importance to the general public. Monitoring public opinion on the role and importance of forests is important for forest management. The pandemic was a watershed moment, since which there has been a systematic and very rapid increase in public pressure to change the existing model of forest management, expressed as a desire to strengthen the social functions of the forest. Using our research as an example, it is clear that awareness of the need for contact with the forest, with nature, is now high among people, which is expressed both in the increased frequency of visits to forests and in a firm belief in the increased importance of the social functions of the forest. At the same time, however, this research suggests that knowledge of economics, or of environmental protection, may not be sufficient to accept that the forest must perform economic functions that are important for the development of the economy (based on divided public opinion on the decline in the importance of the economic functions of the forest) and to understand the full spectrum of the negative impact of the economy on the natural environment (based on the shared opinion on the relationship between the slowdown of the economy and the

improvement of the condition of the natural environment). Hence, in further research on public preferences for forest ecosystem services, more emphasis should be placed on recognizing the level of environmental awareness, and it is imperative that informal education by foresters emphasizes the economic value of the forest and its importance to the development of the regional and national economy.

**Author Contributions:** Conceptualization, E.J. and M.W.; methodology, E.J. and K.J.; software, J.B., S.Z. and K.U.-B.; validation, J.F.; formal analysis, K.J. and K.U.-B.; investigation, M.W., J.B., K.U.-B. and S.Z.; resources, E.J. and J.F.; data curation, E.J.; writing—original draft preparation, E.J., K.J. and M.W.; writing—review and editing, J.F.; visualization, K.U.-B. and S.Z.; supervision, E.J. All authors have read and agreed to the published version of the manuscript.

**Funding:** This research was financed by the State Forests National Forest Holding (Agreement no EO.271.3.13.2019).

**Institutional Review Board Statement:** Not applicable.

**Data Availability Statement:** Data is unavailable due to privacy restrictions.

**Conflicts of Interest:** The authors declare no conflicts of interest.

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
