# Peer review of "How Did COVID-19 Pandemic Stress Affect Poles’ Views on the Role of the Forest?"

_land, doi:10.3390/land13050656_

Round 1

Reviewer 1 Report

Comments and Suggestions for Authors

Dear Authors!

We have read the results of a well thought-out and focused research.
The relevant literature is presented at the beginning of the paper.
A special case of forest utilisation is presented by the authors' team, which has been justified by a pandemic.
The structure of the study is well structured.
The methods used are appropriate. The evaluation methodology is also well thought out.

Two questions arose while reading the paper:

Does geographic conditions influence the answers? One can read about urban-rural divisions, but does the geographical proximity to forests or the geographical location of the place of residence influence the response?

The second question is what changes have taken place in the shaping of Polish environmental awareness since the data were collected.  What new measures, if any, have been taken in this field in Poland?

The conclusions of the study can be used today, but it would be very helpful if it could be explained in a few words (including in the discussion section) whether there have been any changes in this respect in the last 3 years.

Author Response

We really would like to thank the reviewer for the comments.

Two questions arose while reading the paper:

Does geographic conditions influence the answers? One can read about urban-rural divisions, but does the geographical proximity to forests or the geographical location of the place of residence influence the response?

Answer:

Thank you for this question, it is very inspiring for us. I believe that geographic conditions indeed can be an important predictor of social views, not only on the spectrum of ecosystem services, but also on many other threads and phenomena occurring in space. In the survey, the questionnaire's metric included questions about residence. The results clearly show that the place of residence determines the frequency of visits to the forest, and also influences views on whether it is easier to contract a virus in the forest. We did not examine the sctricte thread related to proximity to the forest in this case. But our previous research on the cultural services of the forest showed that distance from residence to the forest and family situation (children) proved to be statistically insignificant characteristics (at a significance level of p<0.05) for the perception of CH as a very important forest service.

The second question is what changes have taken place in the shaping of Polish environmental awareness since the data were collected.  What new measures, if any, have been taken in this field in Poland?

Answer:

This is a very interesting question, and it is certainly worthwhile, as we pointed out in the manuscript, to monitor public opinion on perceptions of forest ecosystem services. However, this type of research requires raising funds and time. So far, despite the many attempts we have made, we have not received financial support that would allow us to conduct such extensive research again. We don't think there is currently a comparable study of this type in Poland.

The conclusions of the study can be used today, but it would be very helpful if it could be explained in a few words (including in the discussion section) whether there have been any changes in this respect in the last 3 years.

Answer:

Thank you very much for this comment, however, to the best of our knowledge, there have been no studies in the past few years that have addressed similar themes. On the other hand, we can see that changes in views are created by the media, and this has certainly not changed since our research. It is difficult to say whether the environmental awareness of Poles has increased - the Ministry of Environment reports published annually on this aspect do not indicate clear, positive changes in this aspect. Given the public's attitudes toward forest management, it seems that awareness of the fact that the forest, through material goods (e.g., providing wood for the market), contributes significantly to economic growth is very limited.

Reviewer 2 Report

Comments and Suggestions for Authors

Dear authors,

Thank you for the opportunity to read your paper. The topic is very interesting, and I found it with great potential to be published. Please find my comments bellow:

Your research provides important insights into the effects of the COVID-19 epidemic on forest visitation and societal perceptions of forests' functions and importance. The introduction skillfully sets the stage by emphasising the importance of forests and the pandemic's possible impact on human behaviour and attitudes towards nature. The methodology section gives a thorough summary of the study's design and data collection procedures, assuring transparency and repeatability. The results section presents the findings in a systematic and informative manner, with clear explanations of the data and pertinent illustrations to highlight significant themes. The discussion section successfully contextualises the results within the current literature, comparing and contrasting data from other studies to provide a thorough overview.

Now some comments to improve:

1. Provide clearer and more specific objectives at the end of the introduction to guide the reader on what to expect from the study.

2. Include more detail on how the sampling was conducted, such as the criteria used for selecting participants and any potential biases.

3. Provide more integration of the literature throughout the discussion to support and contextualize the findings.

4. Provide a concise summary of the main findings and their implications for future research and policy.

5. Clearly outline the limitations of the study and areas for future research.

Thank you and good luck!

Author Response

We really would like to thank the reviewer for comments.

Now some comments to improve:

  1. Provide clearer and more specific objectives at the end of the introduction to guide the reader on what to expect from the study.

Answer:

Thank you very much for this comment. At the end of the introduction chapter, we presented the general objective of the research being conducted and formulated the research hypotheses. We have now supplemented this message with specific objectives integrally related to the hypotheses presented. We added: “Achieving this goal involved the need for two more specific objectives. One was to determine whether people were more likely to visit forests for recreation as a result of the pandemic. The second specific objective was to show whether societal views about forests, in particular the social functions of the forest, changed under the stress of covida.”

  1. Include more detail on how the sampling was conducted, such as the criteria used for selecting participants and any potential biases.

Answer:

Thank you for this comment, we have completed the sampling information.

Liaison duties were largely responsible for the selection of respondents. Their task was to forward the survey link to people they met in the forest, using recreational facilities, walking in the forest, etc., as well as to people who contacted the forest districts about educational activities or anything else related to their stay in the forest. Each time liaison duties asked about the age of the respondents and the purpose of their visit. The questionnaire was completed only by adults over the age of 18 who were interested in recreation in the forest.

  1. Provide more integration of the literature throughout the discussion to support and contextualize the findings.

Answer:

Thank you for your attention, we would like to point out that we have made every effort to ensure that the discussion is integrated into the various threads studied and responds to the general objective as well as the specific objectives and the hypotheses adopted.

  1. Provide a concise summary of the main findings and their implications for future research and policy.

Answer:

Thank you for this comment, we have supplemented the summary with information on implications for future research and policy.

Hence, in further research on public preferences for forest ecosystem services, more emphasis should be placed on recognizing the level of environmental awareness, and it is imperative that informal education by foresters emphasize the economic value of the forest and its importance to the development of the regional and national economy.

  1. Clearly outline the limitations of the study and areas for future research.

Answer:

Thank you for this comment. We signaled the area of future research in the summary. On the other hand, since the research was conducted during a specific period (pandemic), it is difficult to find limitations that could be addressed in the future. We are aware that certainly more control would be useful when it comes to the selection of respondents. However, this element is difficult to achieve with regard to an online survey.